# Survival of Filamentous Cyanobacteria Through Martian ISRU: Combined Effects of Desiccation and UV-B Radiation

**DOI:** 10.3390/microorganisms13051083

**Published:** 2025-05-07

**Authors:** Miguel Arribas Tiemblo, Inês P. E. Macário, Antonio Tornero, Ana Yáñez, Slavka Andrejkovičová, Felipe Gómez

**Affiliations:** 1Centro de Astrobiología (CAB, CSIC-INTA), 28850 Torrejón de Ardoz, Spain; marribas@cab.inta-csic.es (M.A.T.); ipatrunilho@cab.inta-csic.es (I.P.E.M.); ana.yannezv@estudiante.uam.es (A.Y.); 2Geosciences Department, GeoBioTec, University of Aveiro, Campus Universitário de Santiago, 3810-193 Aveiro, Portugal; slavka@ua.pt

**Keywords:** cyanobacteria, Martian regolith simulants, ultraviolet radiation, cyanobacterial growth, cyanobacterial survival

## Abstract

Cyanobacteria are a widespread group of photosynthesizing prokaryotes potentially relevant for space exploration, as they can produce both oxygen and organic matter. These organisms have been repeatedly proposed as tools for colonizing planetary bodies in the solar system. We used several Martian regolith simulants to support the growth of three widespread filamentous cyanobacteria (*Desmonostoc muscorum* UTAD N213, *Anabaena cylindrica* UTAD A212 and an uncharacterized *Desmonostoc* sp.). All cyanobacteria grew well on the surface of the commercial simulants MGS-1 and MMS-2 and in soluble extracts obtained from them, suggesting that these Martian regolith analogs contain everything necessary to sustain cyanobacterial growth, at least in the short term. We also evaluated the survival of the two *Desmonostoc* species under desiccation and UV-B radiation, using the same regolith simulants and two clays: Montmorillonite and nontronite. Desiccation hindered growth, but both cyanobacteria were able to recover in less than 30 days in all cases after desiccation. Short irradiation times (up to 1000 kJ/m^2^) did not consistently affect survival, but longer ones (24,000 kJ/m^2^) could fully sterilize some samples, although cyanobacteria within MGS-1, montmorillonite and nontronite showed signs of recovery in the long term (>70 days). Clays led to very fast recoveries, particularly montmorillonite.

## 1. Introduction

Since the arrival of Apollo 11 on the Moon in 1969 and the establishment of the International Space Station (ISS) in 2000, humanity’s next major challenge has been sending crewed missions to another planet. Mars, being the closest planet to Earth, has been the target of numerous unmanned missions for decades. These include orbiters like Mariner 4 and 9 and Mars Express, as well as landers such as the Viking probes and, more recently, rovers like Curiosity and Perseverance. These missions have significantly enhanced our understanding of the Red Planet, revealing the composition of its crust, its tenuous atmosphere and, notably, evidence of both present and past water through geological studies [1,2].

Organic molecules like chlorobenzene have been detected in the Martian surface, and although they do not necessarily require a biotic origin, they imply that aromatic hydrocarbon and other biomarkers can be preserved within the Martian surface [3]. Although not yet described on Mars, other molecules like lipids [4,5], amino acids [6,7,8,9] and biomarkers like carotenoids [10,11] can be preserved within the first few meters of the Martian surface for very long (>100 Ma) periods of time, which clashes with the idea of Mars as an unequivocally hostile environment for life.

Solar radiation, elements that are essential for life on Earth (CHNOPS) and nutrients like iron, magnesium, nickel and zinc are present within Mars’ atmosphere and rocks [1,12]. Recently, the Perseverance rover discovered sedimentary rocks within Jezero Crater that seem to have formed through the action of liquid water millions of years ago [13]. These and many other findings over recent decades position Mars as a prime candidate for having supported life in the past. Consequently, since the early 21st century, some space agencies have set sights on sending crewed missions to Mars.

Crewed missions of this magnitude entail a series of logistical issues that need to be approached carefully. The most significant challenge is ensuring the supply and survival of astronauts while on the planet’s surface [14]. Current missions to the ISS can be resupplied relatively easily due to its low Earth orbit [15]. However, future missions to the Moon and, particularly, to Mars cannot rely on periodic resupply from Earth, demanding self-sufficiency and the ability to produce food and other goods in situ, a concept known as in situ resource utilization (ISRU) [14,15,16]. This self-sufficiency and independence from Earth are critical not only because of the vast distance between the planets. The economic cost of sending the necessary food for missions, which may last for years, the communication delay of up to 20 min and the uncertainty surrounding the success of rocket launches with supplies make this self-sufficiency imperative for the mission’s success and astronaut safety [16].

Although the ease of use, simplicity and scalability of microorganisms and, in particular, cyanobacteria make them good candidates for the design of biological support systems on other planets [17], not all microorganisms can survive space travel or utilize Martian resources for survival. Extremophiles may be particularly relevant in these extreme conditions, as they exhibit high adaptability to those harsh conditions, such as high radiation or low pressure, which are characteristic of such missions, and they may even be candidates for terraforming Mars [18]. Cyanobacteria have been extensively studied in the context of space exploration and are prime targets for the production of organic matter and oxygen, primarily due to their simplicity, low nutrient requirements, ease of cultivation and their ability to adapt and thrive in a wide variety of ecosystems and extreme environments, from freshwater and saline environments to jungles and deserts and even subterranean conditions [19,20,21].

Cyanobacteria are also the only prokaryotes capable of performing oxygenic photosynthesis, making them invaluable for space exploration. Additionally, their broad range of applications enhances their relevance in this field. On Earth, cyanobacterial oxygen production is substantial, with marine cyanobacteria of the genus *Prochlorococcus* alone accounting for over 13% of photosynthesis in oligotrophic ocean environments [22]. Other cyanobacterial genera like *Synechococcus* also make up a significant fraction of oceanic photosynthesis and are more widespread than *Prochlorococcus* [23,24]. This photosynthetic capacity surpasses that of higher plants. Also, compared to higher plants, cyanobacteria are far less nutritionally demanding and require less space. This leads to higher efficiency in the production of both oxygen and organic matter, which are invaluable resources for maintaining crewed and isolated missions in space [19,25].

Besides oxygen, cyanobacteria can produce other valuable compounds such as amino acids, polysaccharides, lipids and pigments, which have numerous applications in the pharmaceutical and food industries [26]. Some cyanobacteria can be utilized directly as food. In fact, in countries like China and Peru, cyanobacteria from the genera *Nostoc* and *Arthrospira* have been consumed as nutrient-rich food sources including vitamins and proteins for thousands of years [27,28]. *Arthrospira platensis* and *Chlorella vulgaris* are considered food-grade items under EU regulation and are widespread supplements. Other relevant applications of these microorganisms for space exploration include their capacity to produce third- and fourth-generation biofuels [29] or the use of species like *Anabaena* sp., among others, as biofertilizers [19,30].

Besides their applications, cyanobacteria also exhibit high resistance to several relevant stressors such as radiation. Ultraviolet radiation can disrupt biochemical processes in cells and damage DNA, leading to cell death [31]. However, cyanobacteria can produce specific compounds to protect their cellular machinery and DNA from the harmful effects of UV-A (320–400 nm), UV-B (180–320 nm) and UV-C (100–250 nm) radiation [32,33]. Furthermore, genera like *Chroococcidiopsis* exhibit remarkable resistance to desiccation, radiation and microgravity conditions [18,34,35], making these bacteria excellent candidates for space missions.

Following the main purpose of ISRU, the Martian regolith appears as a promising source of essential elements. To this end, a thorough study of Martian soil composition is essential. However, the problems of the limited amount of Martian material available on Earth (from meteorites) and the high costs associated with missions to the planet are somewhat avoided by using terrestrial analog regions [36]. Such regions, including certain areas of the Atacama Desert in Chile (27°22′00″ S, 70°19′56″ W) and the Río Tinto region in Huelva, Spain (37°42′11″ N, 6°33′10″ W), are considered significant analogs for Mars, mainly due to their aridity and high UV radiation levels (in the case of Atacama) or the similarity of their mineral composition (in the case of Río Tinto) [37,38].

In addition to these natural areas, commercial regoliths analogous to Martian rock, such as Mars Global Simulant (MGS-1) [39] or Mojave Mars Simulant (MMS-1/MMS-2) [40], are also available. These regoliths can be supplemented with perchlorates, which have been confirmed to be present on the planet’s surface by missions like Viking, Curiosity and Perseverance [41,42,43,44]. As most species exhibit tolerance to certain concentrations of perchlorates, cyanobacteria might be better suited for growth in the Martian regolith than other photosynthesizing organisms. However, perchlorates have shown a degree of inhibition in the growth of some cyanobacteria [45].

Growth experiments with simulants like MGS-1 have demonstrated growth of *D. muscorum* and *A. cylindrica* at various concentrations of this regolith [16,19]. These results are promising for cyanobacteria, indicating that they could adapt and thrive in Martian soil under atmospheric conditions like those on Earth. The use of microorganisms also presents certain risks, particularly regarding forward contamination of Mars [15,18]. Although many cyanobacteria can produce toxins that are detrimental for human health, those used in this study do not possess that ability [46]. In this article, we aim to examine the potential of selected safe cyanobacteria to grow purely off the Martian regolith simulants. We also assess the resistance of these cyanobacteria to deleterious conditions like UV radiation and desiccation while in the same regolith simulants.

## 2. Materials and Methods

### 2.1. Cyanobacteria Culturing and Identification

Throughout this work we used three widespread filamentous cyanobacteria (*Desmonostoc muscorum* UTAD N213, *Anabaena cylindrica* UTAD A212 and an uncharacterized *Desmonostoc* sp.) as models for the growth of filamentous freshwater cyanobacteria in Martian-relevant mineral substrates. All three cyanobacteria were isolated from freshwater habitats [47] and stored at the Biology Department in the University of Aveiro and later taken to the Spanish Center for Astrobiology (CAB), where the rest of the experimentation took place. These cyanobacteria were maintained in cultures in Synthetic Woods Hole growth medium with periodic culture transfers every month. Cultures were grown under 10 µmol m^−2^ s^−1^ continuous light intensity under no agitation. DNA was extracted from all three cultures. Identification was based on amplification of 16S rRNA through the 27F/1492R bacterial primer pair and CYA 359F/781R, which is specific to cyanobacteria. The first two cultures were sourced from direct reculturing of previously isolated and described species [47]. The evolutionary history was inferred using the neighbor-joining method [48]. The evolutionary distances were computed using the maximum composite likelihood method [49] and are in the units of the number of base substitutions per site. This analysis involved 12 nucleotide sequences. All positions with less than 95% site coverage were eliminated, i.e., fewer than 5% alignment gaps, missing data and ambiguous bases were allowed at any position (partial deletion option). There were a total of 186 positions in the final dataset. Evolutionary analyses were conducted in MEGA11 [50].

### 2.2. Measurement of Cyanobacterial Fluorescence

The growth of each cyanobacterium was monitored through the measurement of the fluorescence intensity measurements using a TECAN Infinite^®^ M Nano + plate reader (Männedorf, Switzerland) Measurements were programmed with 20 s of prior shaking in orbital mode with a 1 mm amplitude and a frequency of 432 rpm. After shaking, the cyanobacteria were excited at a wavelength of 620 nm and the emission intensity was measured at 685 nm, with 4 measurements taken for each well. The absorption at 620 nm corresponds to the maximum absorption peak for phycocyanin, the principal cyanobacterial-specific pigment in the antennae of blue-green cyanobacteria like *Nostoc* [23,51]. The results were transferred to an Excel spreadsheet for further analysis and subsequent graphing using R version 4.3.2.

### 2.3. Martian Regolith Simulants

The two commercially available Martian regolith simulants we used, MGS-1 and MMS-2, were respectively sourced from their commercial providers (Exolith Labs for MGS-1, and The Martian Garden for MMS-2). Polvo de Río Tinto (PRT) or Rio Tinto Dust was made up of rock dust sourced from “Gossan” rocks from the Río Tinto area, rich in iron oxides like hematite and jarosite and considered analogous to those found in some regions of Mars [52]. The mineralogical compositions of MGS-1 and MMS-2 are freely available from their suppliers. PRT was analyzed through X-ray diffraction and was mainly composed of quartz and two iron oxides (hematite and jarosite). For the second stage of the project, we used two clays: Nontronite (NG-1, The Clay Minerals Society, Source Clays Repository, Purdue University) and montmorillonite (STx-1b, The Clay Minerals Society, Source Clays Repository, Purdue University).

### 2.4. Oxidative Capacity of the Regoliths

The oxidative capacity of Martian regolith simulants MGS-1, MMS-2 and PRT was measured using a colorimetric analysis based on the consumption of 1,4-dithiothreitol (DTT). A total of 20 mg of each regolith was weighed into 1.5 mL Eppendorf tubes, making 8 replicates for each regolith. To each Eppendorf 200 µL of a 0.1 M potassium phosphate buffer at pH 7.4 containing 100 µM DTT was added and mixed for 60 min at 37 °C in an Eppendorf ThermoMixer comfort. After incubation, 20 µL (10%) of trichloroacetic acid (TCA) was added to stop the reaction, and samples were centrifuged at 12,000 rpm for 10 min at room temperature in an Eppendorf Centrifuge 5415 R.

After centrifugation, the samples were allowed to rest for 15 min to sediment the regolith. A total of 100 µL of the supernatant was extracted and transferred to transparent 96-well plates. To each well, 200 µL of a 0.4 M Tris-HCl solution containing 20 µM ethylenediaminetetraacetic acid (EDTA) and 5 µL of a 10 mM Ellman’s reagent (DTNB) dilution were added in the dark, leading to the immediate development of a yellow coloration. The positive control was performed in the absence of any regolith simulant. The absorbance (AB) of each well was measured at 412 nm. The calculation for the DTT consumption ratio was performed using the following equation:$$\mathrm{Consumption}\;\mathrm{ratio}=\frac{\left({AB}_{c+}\right)-({AB}_{rego})}{\left(A_{c+}\right)}$$

More oxidizing substrates lead to higher DTT consumptions and lighter yellow colors.

### 2.5. Preparation of the Soluble Fraction of Martian Regolith Simulants

Soluble fractions were prepared using MGS-1, MMS-2 and PRT. First, 1 g of each simulant was weighed and added to 5 mL of either distilled ultrapure water (dH_2_O) or MBL growth medium in 15 mL Falcon tubes, vortexed for one hour and centrifuged at 9000 rpm for 10 min at room temperature in an Eppendorf™ Centrifuge 5810 R. The soluble fraction was then sterilized in an autoclave and stored.

### 2.6. Cyanobacteria Growth Over Martian Regolith Simulants and in Their Soluble Fraction

Growth was carried out in Thermo Scientific 96 Well White Bottom plates. Each cyanobacterium (*D. muscorum, A. cylindrica* and *Desmonostoc* sp.) was grown on a separate plate. As growth was carried out in both dH_2_O and MBL, each plate was further divided into two sections. Then, 1.5 mL of each grown culture of cyanobacteria was centrifuged at 5000 rcf for 10 min; the supernatant was discarded, and the cyanobacterial pellet was resuspended in 5 mL of either dH_2_O or fresh growth medium in order to start the experiments with an optical density (OD 440 nm) of 1.

#### 2.6.1. Growth in the Soluble Fraction of 3 Martian Regolith Simulants

One hundred and eighty microliters of the soluble fraction obtained from MGS-1, MMS-2 or PRT was added to the corresponding wells. Six replicates of each condition were made following vertical columns in the plate. Distilled water (200 µL) was added to the wells along the edges of the plate to prevent evaporation of the extracts during the growth period. Twenty microliters of the corresponding cyanobacteria (in growth medium or dH_2_O) was added to the soluble fraction, leading to an overall 10× dilution.

In all cases, controls were added to assess growth in the absence of any soluble fraction, in either growth medium or dH_2_O alone. Plates were then covered with the lid and sealed with Parafilm and gently agitated for 5 min to homogenize the cyanobacteria with the soluble fractions on a Heidolph Titramax 1000 platform shaker. All plates were left to grow under 10 µmol m^−2^ s^−1^ of continuous polychromatic artificial light for a total of 70 days at room temperature, with fluorescence measurements taken every 3 days during the early stages of growth and weekly once growth stabilized.

#### 2.6.2. Growth over Three Regolith Simulants

Twenty milligrams of each sterilized regolith simulant (MGS-1, MMS-2 and PRT) was weighed and placed in the wells following the same distribution as in the soluble fraction plates (Section 2.6.1). Then, 20 µL of cyanobacteria (in growth medium or dH_2_O) and 180 µL of growth medium or dH_2_O were added to the wells to allow for the same 10× dilution as in the case of the soluble fractions. Controls were performed similarly, growing each cyanobacterium in sea sand previously washed three times with distilled water to provide a mineral substrate devoid of any potentially beneficial soluble fraction.

### 2.7. Cyanobacteria Growth in Regolith Under Stress

#### 2.7.1. Drying and UV Exposure

Cyanobacteria were similarly plated prior to desiccation and irradiation. Only *D. muscorum* and *Desmonostoc* sp. were used in these experiments, as *A. cylindrica*’s growth led to regolith aggregation and would not allow for replicable results. Cyanobacteria were pelleted and concentrated 1.6 times in dH_2_O. These concentrates were then added to 20 mg of each substrate (MGS-1, MMS-2, PRT, montmorillonite, nontronite and an empty control) in quintuplicate and gently agitated for 10 min to allow for complete homogenization. Plates were then vacuum dried in a Speedvac vacuum centrifuge for 30 min, which fully dried all samples.

Plates were next placed in an Opsytec Dr Gröbel BS-02 UV-B irradiation chamber. The chamber is equipped for irradiation in the UV-B region, with an estimated fluence in the 250–350 nm range of 92 W/m^2^, and is equipped to maintain a working temperature range of 25–30 °C. Samples were exposed to 7 irradiation periods: 0 min, 1 min, 10 min, 60 min, 120 min, 240 min and 72 h. These are equivalent to doses of 0, 5.5, 55, 330, 660, 1300 and 24,000 kJ/m^2^ in the 280 to 315 nm range. The daily fluence of UV-B radiation on Mars is estimated at around 361 kJ/m^2^ [53], from which we derive that 1 h of irradiation equals around 1 sol of Martian UV-B irradiation, dose-wise.

Following irradiation, dried plates were rehydrated in 200 µL of growth medium, which led to an accumulated 3x dilution from the original culture. Three times more cyanobacteria were used in this case, as desiccation was expected to heavily reduce cell viability. Two hundred microliters of dH_2_O was added to the peripheral wells of the plate to prevent evaporation of the growth medium. Plates were sealed with Parafilm and left under 10 µmol m^−2^ s^−1^ of continuous polychromatic visible light for 67 days at room temperature, with fluorescence measurements taken every 3 days during the early growth stages and weekly once growth stabilized.

#### 2.7.2. UV Exposure in Regolith Simulants in the Presence of Water

The protocol was similar for non-desiccated samples. *D. muscorum* and *Desmonostoc* sp. were similarly concentrated. First, 40 µL of each concentrate was added to 20 mg of each substrate (MGS-1, MMS-2, PRT, montmorillonite, nontronite and an empty control). Samples were then irradiated for the designated periods (0 min, 1 min, 10 min, 60 min, 120 min, 180 min and 240 min). Samples were always kept moist and immediately resuspended in 200 µL of growth medium following exposure. Two hundred microliters of dH_2_O was added to the peripheral wells of the plate to prevent evaporation of the growth medium. Plates were sealed with Parafilm and left under 10 µmol m^−2^ s^−1^ of continuous polychromatic visible light for 35 days at room temperature, with fluorescence measurements taken every 3 days during the early growth stages and weekly once growth stabilized.

### 2.8. Curve Modeling Using Growthcurver

*Growthcurver* is an R package designed to efficiently fit growth curve data to the standard form of the logistic equation.$$N_{t}=\frac{K}{1+\left(\frac{K-N_{0}}{N_{0}}\right)e^{-rt}}$$
This equation aims to describe the growth of microorganisms and is often used with measures of optical density (OD) or fluorescence. From each model we derive a series of parameters (the growth rate (r, days^−1^), the initial population size (N_0_) and the carrying capacity (K)), which provide meaningful information with biological relevance.

## 3. Results

### 3.1. Determination of the Oxidative Pressure Exerted by the Regolith Simulants

The extent of the oxidative damage of each Martian regolith simulant was analyzed through their ability to oxidize a heavily reducing agent like dithiothreitol (DTT). The consumption of this molecule was measured through a colorimetric assay. Data in Table 1 show that PRT was, as expected, the more oxidative substrate, followed by MMS-2. MGS-1, the least iron-oxide-rich simulant, displayed the lowest oxidative capacity.

### 3.2. Cyanobacterial Growth in Martian Regolith Simulants

#### 3.2.1. Identification of the Cyanobacterial Strains

The first isolate (Figure 1A) displayed over 98% identity (27F/1492R) with several *Desmonostoc* species and was confirmed to be *Nostoc muscorum* (UTAD_N213), now reclassified as *Desmonostoc muscorum*. The second culture (Figure 1C) had over 99% percentage identity (27F/1492R) with *Anabaena cylindrica* (PCC 7122) and was also confirmed to be *Anabaena cylindrica* (UTAD_A212) based on morphological analysis. Both strains had been previously isolated and cultured from Portuguese rice fields. The last strain (Figure 1B) was isolated more recently and was harder to identify. Sequencing based on CYA 359F/781R showed >90% identity with several Desmonostoc species, and it is hereafter referred to as *Desmonostoc* sp.

A phylogenetic tree (Figure 1D) was constructed using the three strains, several closely related microorganisms and another two, more distant cyanobacteria (*Sinocapsa zengkensis* and *Chroococcidiopsis cubana*) to anchor the tree. Strains 1 (*Desmonostoc muscorum* (UTAD_N213)) and 3 (*Desmonostoc* sp.) were very closely related, their closest relative being *Desmonostoc punense*. Strain 2 (*Anabaena cylindrica* (UTAD_A212)) was closest to another *Anabaena cylindrica*. Strains 1 and 2 were directly recultivated from isolated cultures [47] and the naming conventions established in the describing article were maintained throughout the study.

#### 3.2.2. Growth in the Soluble Fraction of Martian Regolith Simulants

The first aim was to determine whether the soluble material present in each of the three Martian regolith simulants (MGS-1, MMS-2 and PRT) was enough to sustain the growth of cyanobacteria (which we assayed through growth in dH_2_O) and if any oxidative or damaging compounds could prevent or hinder that same growth (which we assayed through growth in growth medium). Growth in pure dH_2_O did not allow for growth in any of the three cyanobacteria, and fluorescence slowly declined as cells died and phycocyanin degraded (Figure 2).

None of the *Desmonostoc* species grew in the soluble phase of PRT, but *A. cylindrica* managed to do so to some degree after a 20-day period of adaptation. MGS-1 successfully supported growth in all three cyanobacteria and both in growth medium and in dH_2_O, and no strong differences were observed between them, which suggests that MGS-1 can successfully allow for the growth of the cyanobacteria through soluble compounds present within it, and contains no soluble material capable of inhibiting or hindering growth. MMS-2 also allowed for growth in all cases, but growth in dH_2_O was significantly slower in the two *Desmonostoc* species. Growth in *A. cylindrica* was similar in the two cases, without significant variation between MGS-1 and MMS-2.

#### 3.2.3. Growth over Martian Regolith Simulants

Following growth in soluble phases, all three cyanobacteria were cultured while in direct contact with each regolith simulant, in both growth medium and dH_2_O. The controls were carried out in sterilized sea sand (Figure 3) as the mineral substrate, so that cyanobacteria had a surface to attach to. Unlike in the previous case, both *A. cylindrica* and *Desmonostoc* sp. grew well in dH_2_O, which implies they managed to obtain enough necessary nutrients from the washed sea sand or that the sand allowed for the formation of self-sustaining niches. *D. muscorum*, on the other hand, did not grow in dH_2_O.

*A. cylindrica* managed to grow after approximately 30 days of acclimation both in growth medium and dH_2_O while in contact with PRT. The peak in growth was, as in the case of the soluble extracts, well below all the other conditions, and the peak was reached around the 45-day mark, 15 days later than in the soluble fraction extracts. *D. muscorum* did not grow under PRT in this case either, but *Desmonostoc* sp. managed to do so in growth medium, which further suggests it may be a more resilient species. All three cyanobacteria grew well in MGS-1 and MMS-2 (Figure 3A). *A. cylindrica* showed no significant differences in growth in MGS-1, as the growth curves in growth medium and dH_2_O are identical. This was not the case in MMS-2, as once again growth-medium-supplemented cultures grew better, which further suggests that MMS-2 either lacks enough nutrients or, more likely, can hinder the growth in some cases through oxidation.

In both *Desmonostoc* species MGS-1 and MMS-2 behaved similarly, with MGS-1 leading to slightly faster growth than MMS-2. Cultures grown in growth medium also grew faster than those in dH_2_O in both cases, although these differences were not as noticeable in *Desmonostoc* sp., which further cements it as the more resilient of the two. Parametric analysis (Figure 3B) further supports this. MMS-2 consistently displayed higher inflection points and lower growth rates, both consistent with subpar growth. This was more noticeable in dH_2_O than in MBL.

### 3.3. Cyanobacterial Survival Under Deleterious Martian Surface Conditions

Although *A. cylindrica* showed promising results in the previous section, its propensity to form thick (>0.1 mm) clusters of filaments meant it could not be effectively homogenized with the mineral substrates. It also meant that any irradiation or desiccation experiments would not be easily reproducible, as it causes interference with fluorescence measurements. Because of this, the next section focuses solely on the two *Desmonostoc* species, which grew in a more uniform suspension and could be easily partitioned. We aimed to study the survival of these two cyanobacteria under a set of environmental stresses all too common in the Martian surface: desiccation and UV radiation.

#### 3.3.1. Survival and Growth After UV-B Irradiation and Desiccation

In this set of experiments, we subjected both *Desmonostoc* species to extreme desiccation through a vacuum centrifuge, followed by six doses of UV-B radiation, from 1 min (5.5 kJ/m^2^ or the equivalent of 0.017 Martian sols) to 72 h (24,000 kJ/m^2^ or 72 Martian sols). The first ten days of the growth profiles of both cyanobacteria (Figure 4) are dominated by a steady decline in fluorescence intensity, associated to the death and degradation of most of the initial population of cyanobacteria.

The growth profiles of *D. muscorum* (Figure 4A) and *Desmonostoc* sp. (Figure 4B) are very similar. No growth was observed in PRT. It was also noticeable that low doses of UV-B radiation (up to 1300 kJ/m^2^) did not heavily affect growth, although a clear trend could be observed in which 1, 2 and 4 Martian sols of UV-B irradiation did tend to slow the rate of growth in most cases. However, higher doses (24,000 kJ/m^2^ or 72 dose-equivalent Martian sols) did lead to significantly more damage and very heavily impaired growth in all samples, although in some cases a very slow recovery could be observed in some wells after the 40-day mark.

Both *Desmonostoc* species grew well in MGS-1 following desiccation and irradiation, only showing slightly slowed growth at the 24,000 kJ/m^2^ dose, particularly in the case of *D. muscorum*. The other substrates did not exhibit a nearly as strong protective effect (for the 24,000 kJ/m^2^ dose), but both clays (montmorillonite and nontronite) show some degree of growth in both cyanobacteria in the long term (>40 days), as well as *Desmonostoc* sp. in MMS-2. In general, *Desmonostoc* sp. appeared to recover faster than *D. muscorum*, although the differences were not very noticeable.

The curves in Figure 4 were modeled using *Growthcurver*, which fitted the five replicate curves for each condition to a logistic regression. From the model curve, a set of five parameters was retrieved: K (carrying capacity), n_0_ (initial population), r (growth rate), t_mid (time at which the population reaches half of the carrying capacity) and t_gen (generation time). The relationship between r and t_mid was used to infer the impact of the desiccation and radiation in the two *Desmonostoc* species. Figure 5A,B show the inflection point (t_mid) after each dose for *D. muscorum* and *Desmonostoc* sp., respectively. In most cases, there is very little difference in the inflection point at lower doses (up to 660 kJ/m^2^), although a clear exception was *Desmonostoc* sp. in the absence of any substrate, where a steady increase in inflection point can be observed.

While no significant growth could be observed in *D. muscorum* following a dose of 24,000 kJ/m^2^ (with the exception of MGS-1), *Desmonostoc* sp. displays consistent, although significantly higher, inflection points in all substrates but MMS-2, which suggests it may be more resistant to desiccation and UV-B-induced damage, particularly within the tested conditions. Plotting the growth rate (r) and inflection point (t_mid) of each condition together led to a two-dimensional plot (Figure 5C) in which several groups rose. Montmorillonite consistently displayed the higher r and lowest t_mid, both pointing to fast growth. Nontronite then showed lower growth rates and slightly longer inflection points. Both MGS-1 and MMS-2 behaved similarly, with lower growth rates and longer inflection points than the clays. Controls showed the slowest growth rates, but smaller inflection points than MGS-1 or MMS-2. *Desmonostoc* sp. reached the inflection point 5 days before than *D. muscorum* on average.

#### 3.3.2. Survival and Growth After UV-B Irradiation in a Humid Environment

In the next set of experiments, we subjected both *Desmonostoc* species to another six doses of UV-B radiation, from 1 min (5.5 kJ/m^2^ or the equivalent of 0.017 Martian sols) to 4 h (1320 kJ/m^2^ or 4 Martian sols). This was all performed in the presence of a thin (<1 mm) film of water, which prevented desiccation during irradiation. Due to evaporation during the exposure, 4 h was the upper limit to prevent sample desiccation. The growth curves for *D. muscorum* (Figure 6A) and *Desmonostoc* sp. (Figure 6B) were very similar in both cases. Irradiation appeared to have an even lesser effect in this case, although some effects were clear.

No significant differences in growth were observed in MGS-1 in these conditions, implying that again MGS-1 was the substrate that best protected the cells from radiation. Both cyanobacteria grew comparably worse in MMS-2, which may be related to an interaction between its oxidative soluble fraction and UV radiation. Growth in montmorillonite was not heavily affected by smaller doses of radiation. Heavy irradiation on nontronite slowed growth in both cyanobacteria. PRT again prevented any growth, owing to its acidity. In the absence of any mineral substrate, cyanobacteria unintuitively displayed enhanced growth at higher doses. This abnormal growth also prevented them from being modeled with *Growthcurver.*

Following modeling with *Growthcurver*, we evaluated how changes in the inflection point of *D. muscorum* (Figure 7A) and *Desmonostoc* sp. (Figure 7C) cultures varied following irradiation. MGS-1 showed no differences in t_mid at any dose, which implies it was capable of effectively protecting cyanobacteria within it. MMS-2, on the other hand, showed a significant increase in t_mid at doses over 660 kJ/m^2^. The two cyanobacteria behaved differently in montmorillonite, as *D. muscorum* grew significantly slower at doses over 990 kJ/m^2^, which was not observed in *Desmonostoc* sp. Nontronite only showed slowed growth after a dose of 1320 kJ/m^2^. Unprotected controls and PRT could not be modeled as they did not follow a logistic regression.

*Desmonostoc* sp. (Figure 7D) showed increased resistance to desiccation across all substrates when compared to *D. muscorum* (Figure 7B). Desiccation led to longer inflection points (by 10 to 25 days) and to reduced growth rates (by 0.05 to 0.2). The four mineral substrates used also heavily affected growth recovery. In all cases montmorillonite-grown cyanobacteria displayed high growth rates and very short inflection points. Nontronite appeared as the second-best performer in dry samples, particularly in *Desmonostoc* sp. This effect was not as noticeable in wet samples, as the data for nontronite and MGS-1 appear very close, to the point of not being distinguishable. MMS-2 underperformed compared to the other three substrates in the presence of water, which might be due to its abundance in oxidative compounds that would be at their most damaging if in solution.

#### 3.3.3. Effects of Desiccation on Cyanobacterial Growth

Figure 8 focuses on the effects of desiccation in the absence of UV radiation. Under these circumstances, both cyanobacteria followed a similar pattern. Following desiccation, the time taken to reach the inflection point in most cultures increased 100–400%, and the growth rate dropped 25–50%. Changes in inflection point were more noticeable than in growth rate, suggesting that the changes are mostly caused by cell death and that cell viability following rehydration is not heavily impaired. The overarching patterns shown by each mineral substrate did not significantly change after desiccation, as montmorillonite displayed the fastest recoveries in both wet and dried samples, followed by nontronite, MGS-1 and MMS-2. Nontronite overperformed compared to MGS-1 when dry, likely due to retaining moisture, while MGS-1 overperformed compared to MMS-2 in wet cultures. This may have been caused by the solubilization of oxidative compounds present within MMS-2.

## 4. Discussion

The initial set of experiments (Figure 1 and Figure 2) aimed to determine whether cyanobacteria could be feasibly grown purely from Martian-sourced materials, mainly Martian regolith. Cyanobacteria were capable of sustaining growth in the soluble fraction of two Martian regolith analogs (MGS-1 and MMS-2), while they struggled to do so in PRT, which was mainly composed of acidic rock dust from Río Tinto [35,47]. Both Martian and Lunar regoliths are expected to exhibit pHs close to neutrality, and generally above 6 [54], which suggests that the susceptibility of cyanobacteria to acidic mineral substrates should not be taken as an immediate threat to their use in space exploration.

While cyanobacteria grew better in growth-medium-supplemented MMS-2 extracts than in those made up purely of dH_2_O, this was not the case for MGS-1, where growth in both conditions was undistinguishable. MMS-2 soluble extracts, be it through the absence of essential soluble compounds or through artifactual oxidation, could only allow for subpar growth in the absence of standard growth media. MGS-1, on the other side, is a self-sufficient source of nutrients and can fully support the growth of phototrophs. Colonization of the Martian regolith simulants (Figure 3) paints a similar picture to that of growth in the soluble fraction. When grown purely on dH_2_O and the corresponding simulant, both *Desmonostoc* species reached the inflection point significantly faster (>5 days) in MGS-1 than in MMS-2, and these differences practically disappeared when grown on growth medium. *A. cylindrica* did not follow this pattern and formed inhomogeneous aggregates throughout the regolith, which made comparison difficult. This cyanobacterium was also the only one to grow in the very acidic PRT, although at a very reduced rate. Based on all this, sections of the Martian regolith chemically similar to MGS-1 should allow for the growth of cyanobacteria, be it purely in their soluble components or on their surface, as a first step for the biofertilization of Martian soils.

The next section focused on the two *Desmonostoc* species and on the recovery of their viability after being irradiated and desiccated within the different substrates. Desiccated samples took 2–4 times longer to reach levels than undesiccated samples (Figure 8). Low (<1000 kJ/m^2^) UV-B doses had very little influence in the recovery of cyanobacteria. However, more intense doses (up to 24,000 kJ/m^2^) heavily hindered growth, and even prevented growth altogether in some conditions, particularly in the case of MMS-2. A complete absence of growth was only observed in heavily irradiated samples, as all desiccated samples eventually recovered. Short-term UV irradiation does not appear to be a significant issue, but longer periods can fully sterilize the first few millimeters of samples, which desiccation alone cannot.

Of all the mineral substrates, montmorillonite led to the fastest recoveries. This could be due to its ability to form safe niches and to retain water [55], as well as its high reflectance, leading to most of the light being free for the cyanobacteria to absorb, and to the absence of toxic chemicals within it. Nontronite led to slower and less intense growth recoveries than montmorillonite but allowed for faster growth than MGS-1 and MMS-2. Its darker color may lead to less light being directed to the cells, and it may be less effective at retaining water. In both clays, *D. muscorum* and *Desmonostoc* sp. had similar growth rates, but the inflection times of *D. muscorum* were, on average, 5 days longer. This may be caused by a higher susceptibility to desiccation.

In irradiated but not desiccated samples (Figure 5 and Figure 6) most of the previously observed patterns were conserved, although growth was faster, growth rates were higher and inflection points were shorter. Some changes could also be observed, as nontronite and MGS-1 behaved similarly before desiccation, but nontronite clearly outperformed MGS-1 after desiccation, suggesting that its moisture-retaining properties can be relevant in desiccated samples. MGS-1 and MMS-2 also showed similar results following desiccation, but MGS-1 overperformed compared to MMS-2 in solution, likely because of damaging oxidative compounds present within MMS-2. Of the two cyanobacteria, *Desmonostoc* sp. consistently grew faster, which suggests that significant differences may exist even between very genetically close strains.

Throughout the study we used a set of freshwater cyanobacteria which are not extremophilic or particularly resistant to abiotic sources of stress. Their relative success in growing over mineral surfaces similar to those of Mars suggests that more resistant, polyextremophilic cyanobacteria like the thermophilic *Thermosynecococcus* [56] and the desiccation-tolerant *Chroococcidiopsis* [57] are likely to do well as well. However, mesophilic cyanobacteria tend to exhibit the highest growth rates and productivity, and their ease of cultivation makes them priority targets, so long as extremophilic cyanobacteria do not overperform substantially in other areas.

Although filamentous freshwater cyanobacteria from the *Desmonostoc* genus are not amongst the most resilient cyanobacteria, they have developed a variety of mechanisms that confer protection from the sources of stress studied in this article. Akinetes, specialized dormant cells, allow filamentous cyanobacteria to recover from occasional desiccation, but they are unlikely to resist repeated desiccation cycles [58]. These cyanobacteria have also developed mechanisms to resist UV radiation [59] that are comparably more robust than those of some microalgae. These range from the biosynthesis of antioxidants and UV-absorbing compounds like scytonemin to enhanced DNA repair and resynthesis of proteins. *Desmonostoc* species are, however, not expected to be particularly UV-resistant either. The successful growth and recovery of these cyanobacteria in the condition we assayed then suggest that other, slower growing but more resistant and specialized cyanobacteria are likely to be preserved under similar conditions as well.

Desiccation was immediately followed by rehydration under standard growth conditions, which limited the amount of time cyanobacteria were effectively desiccated. The process of desiccation, however, appears to be the main deleterious force, and maintaining that state for prolonged periods of time only slowly reduces viability. Akinetes from other Nostocales species have been reliably germinated from 1800-year-old sediments [60]. Germination rates progressively decrease with time, as recent akinetes exhibit near 100% recovery rates, while akinetes >30-year-old exhibit drops in germination rates to 6–8% [61]. These are, however, still very high germination rates, and cyanobacteria used for ISRU are unlikely to be desiccated for periods much longer than 2 to 3 years. Based on this, so long as cyanobacteria are protected from UV light, they are likely to be preserved within the Martian regolith for years, as the deleterious effects of prolonged desiccation and ionizing radiation should only become relevant after longer (>100 years) exposure times or timeframes too long to be relevant for the ISRU.

## 5. Conclusions

All three cyanobacterial species originally assessed in this work grew well in pure soluble fraction extracts from MGS-1 and MMS-2, two Martian regolith simulants. Unlike MGS-1, cyanobacteria showed grew worse in MMS-2 extracts alone than in those supplemented with MBL, a growth medium for cyanobacteria, implying that its oxidative nature might be hindering growth. Colonization of the solid phase of the same regolith simulants showed similar results. dH_2_O and sterilized MGS-1 and MMS-2 were enough to allow for the full colonization of the regolith’s surface within 70 days. This opens the door to their use as biofertilizers in other terrestrial bodies. MMS-2 did, however, also show reduced growth in this case.

Low radiation doses (<1000 kJ/m^2^, or 4 Martian sols of accumulated UV-B radiation) led to small changes in growth recovery, while higher ones (>24,000 kJ/m^2^, or 72 Martian sols) heavily hindered growth and led to either slow recoveries (>50 days) or complete sterilization of the samples. MGS-1 showed the least variation at all doses and could lead to significant UV attenuation and protection. Desiccation-induced damage led to an immediate loss of viability, unlike UV-induced damage, which is cumulative. Desiccation increased the inflection point of all samples from the 1–10 day range to the 10–40 day range, but all cultures fully recovered from it. Montmorillonite and nontronite, two clays, allowed for the best recoveries in all cases, with montmorillonite leading to extremely fast recoveries. The hygroscopic properties of clays can quickly recover the viability of cyanobacterial cells desiccated within them.

Filamentous cyanobacteria appear to be great candidates for the biofertilization of mineral soils and regoliths on other planetary bodies. MGS-1 and MMS-2, two commercially available Martian regolith simulants, could sustain cyanobacterial growth purely from their intrinsic mineral composition, and the addition of growth media like MBL only slightly improved growth. All three cyanobacteria also efficiently colonized both regolith simulants within a relatively short 70-day timeframe, which posits them as first-round biofertilizers that could prepare Martian and outer planetary regoliths for the growth of more complex plants. The two *Desmonostoc* species we assayed also showed significant resistance to desiccation, being able to recover in less than 40 days in most cases, and to UV-B exposures of up to 4 Martian sols, which hardly affected survival. Exposures over 72 sols should be enough sterilize the surface of most cyanobacteria-containing samples, so strategies for UV attenuation and protection should be devised, because as obligate phototrophs, cyanobacteria can only thrive in direct contact with solar radiation.

## Figures and Tables

**Figure 1 microorganisms-13-01083-f001:**
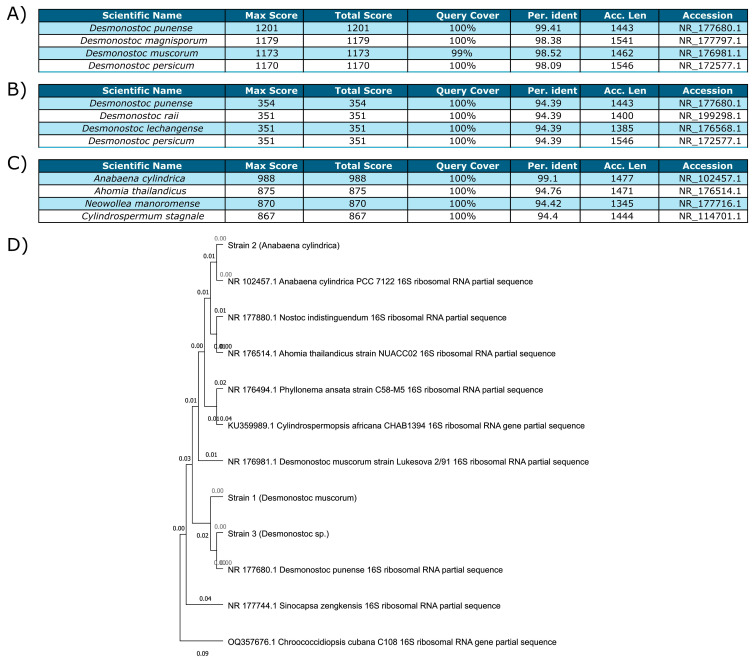
Each table shows the four top results for analysis through BlastN in each strain. Queries were run against the rRNA/ITS database through a discontiguous megablast. (**A**) Strain 1 (*Desmonostoc muscorum* (UTAD_N213)); (**B**) Strain 3 (*Desmonostoc* sp.); (**C**) Strain 2 (*Anabaena cylindrica* (PCC 7122)); (**D**) Maximum likelihood phylogenetic tree of the three strains, their closest relatives and two less closely related anchor species.

**Figure 2 microorganisms-13-01083-f002:**
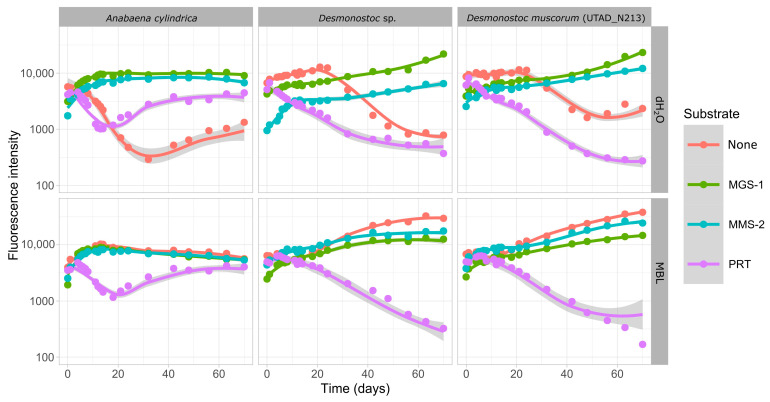
Growth of three cyanobacteria in the soluble fraction of three Martian regolith simulants. Six replicates were prepared for each condition. All points displayed depict the average for each condition. Curves were smoothed using LOESS, a non-parametric method for smoothing. The shaded region depicts the confidence interval. Loess smoothing was necessary as the growth pattern of the cyanobacteria did not fully adjust to a logistic regression.

**Figure 3 microorganisms-13-01083-f003:**
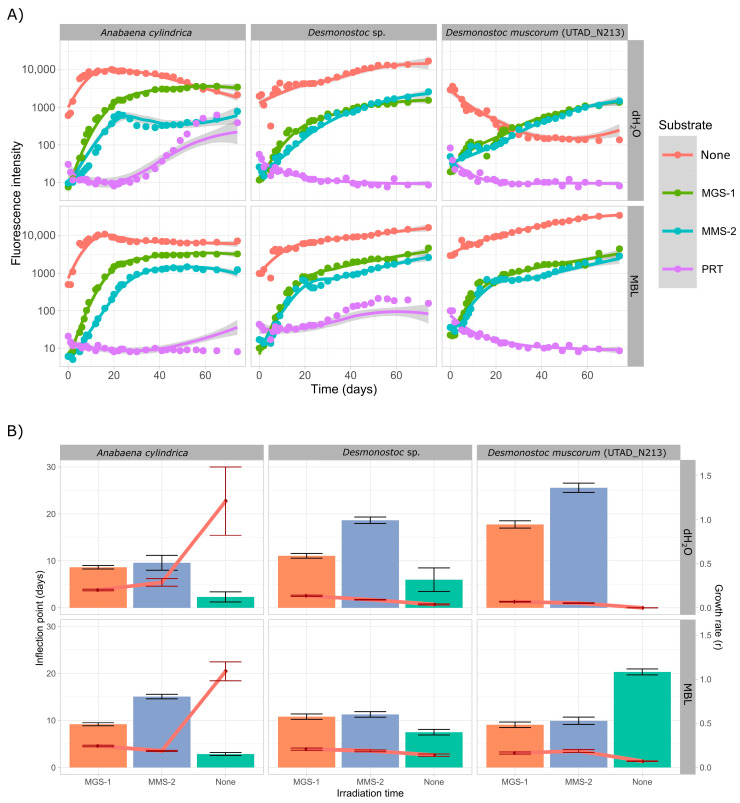
(**A**) Growth of three cyanobacteria on the surface of three Martian regolith simulants. Six replicates were present in each condition. All points displayed depict the average for each condition. Curves were smoothed using LOESS, a non-parametric method for smoothing. The shaded region depicts the confidence interval. (**B**) Representation of the inflection point (t_mid, in days) and the growth rate (r, in days^−1^) of the modeled logistic growth curved obtained through the *Growthcurver* R package. Data for PRT are not shown as it led to inconsistent growth and could not be properly modeled.

**Figure 4 microorganisms-13-01083-f004:**
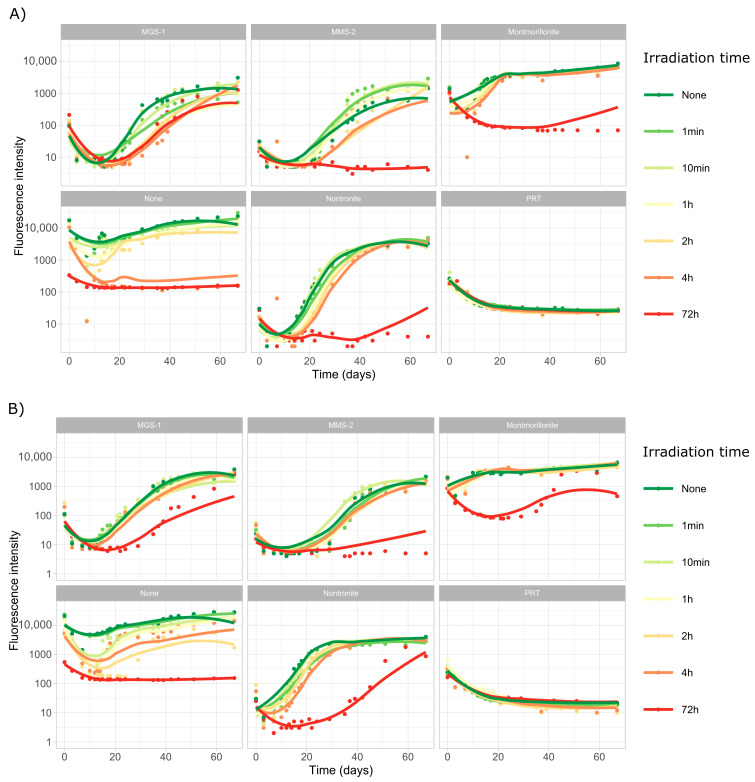
Growth curves of *D. muscorum* (**A**) and *Desmonostoc* sp. (**B**). Each cyanobacterium was grown in MBL medium (using five replicates) after being desiccated and subjected to set amounts of UV-B radiation. Dose is shown in minutes or hours and equals 0, 5.5, 55, 330, 660, 1300 and 24,000 kJ/m^2^ in the 280 to 315 nm range or 0, 0.017, 0.17, 1, 2, 4 and 72 dose-equivalent Martian sols. Curves were smoothed using LOESS, a non-parametric method for smoothing, to account for the many external influences that affected growth, which included the progressive degradation of leftover phycocyanin in the first week of incubation.

**Figure 5 microorganisms-13-01083-f005:**
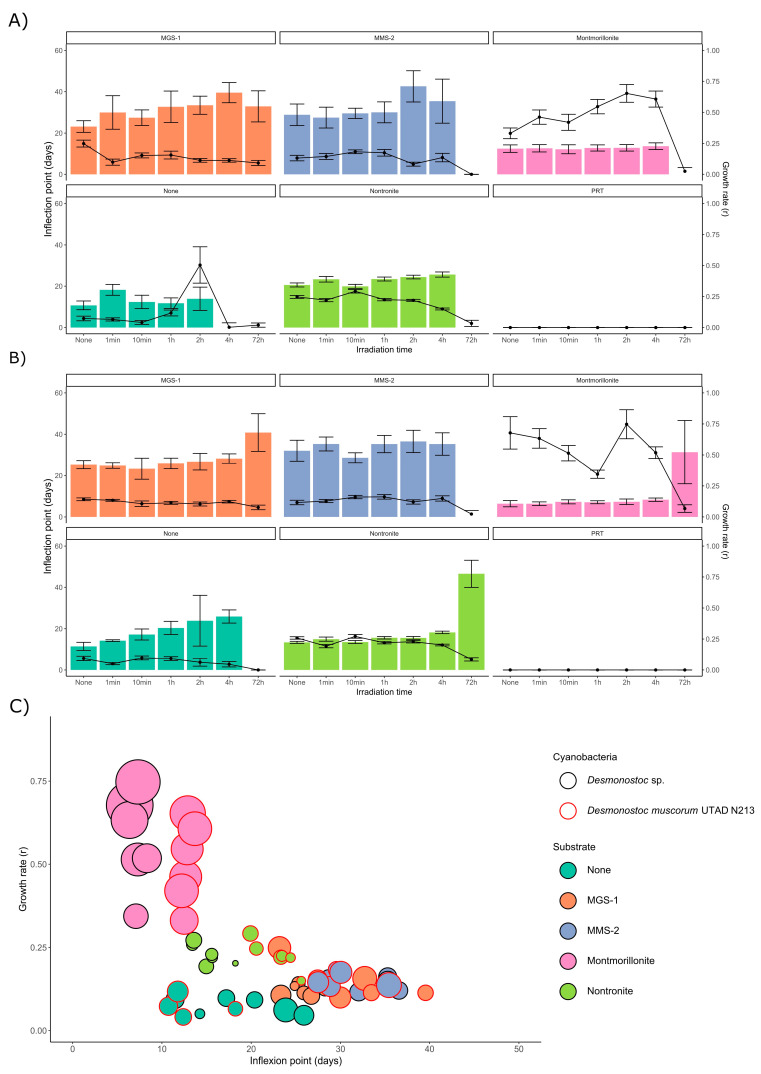
(**A**) Inflection point (t_mid, as bars) and growth rate (r, as points) of the logistic regression obtained from modeling the growth of *D. muscorum* after being desiccated and irradiated in five mineral substrates. (**B**) Equivalent data for *Desmonostoc* sp. (**C**) Representation of the growth rate (r, points) and inflection day (t_mid, bars) of both cyanobacteria in all substrates but PRT (where no growth took place). Each point corresponds to the model derived from five replicates of each cyanobacterium, substrate and dose. As no significant differences took place in t_mid in doses up to 1320 kJ/m^2^, points are not differentiated based on irradiation time. Doses of 24,000 kJ/m^2^ led to either no growth or very high t_mid and are absent from the representation. The radius of each point is the standard error in growth rate (*y*-axis).

**Figure 6 microorganisms-13-01083-f006:**
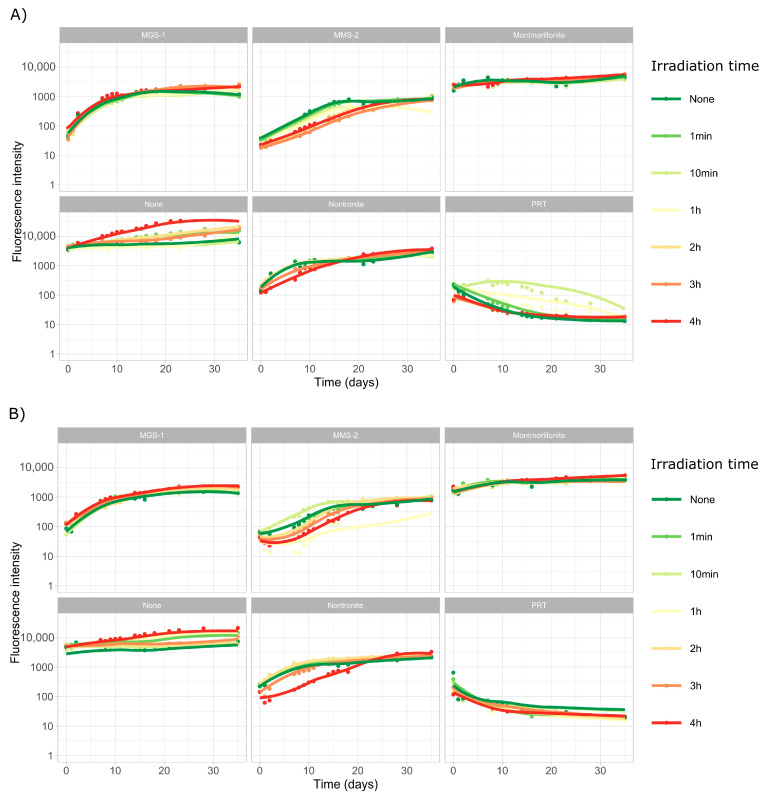
Growth curves of *D. muscorum* (**A**) and *Desmonostoc* sp. (**B**). Each cyanobacterium was grown in MBL medium (using five replicates) after being irradiated for set amounts of time in five mineral substrates while protected from desiccation by a 1 mm film of water. Dose is shown in kJ/m^2^ in the 280 to 315 nm range and equals 0, 5.5, 55, 330, 660, 990,1300 kJ/m^2^ in the 280 to 315 nm range or 0, 0.017, 0.17, 1, 2, 3 and 4 dose-equivalent Martian sols. Curves were smoothed using LOESS, a non-parametric method for smoothing, to account for the many external influences that affected growth, which included the progressive degradation of leftover phycocyanin the first week of incubation.

**Figure 7 microorganisms-13-01083-f007:**
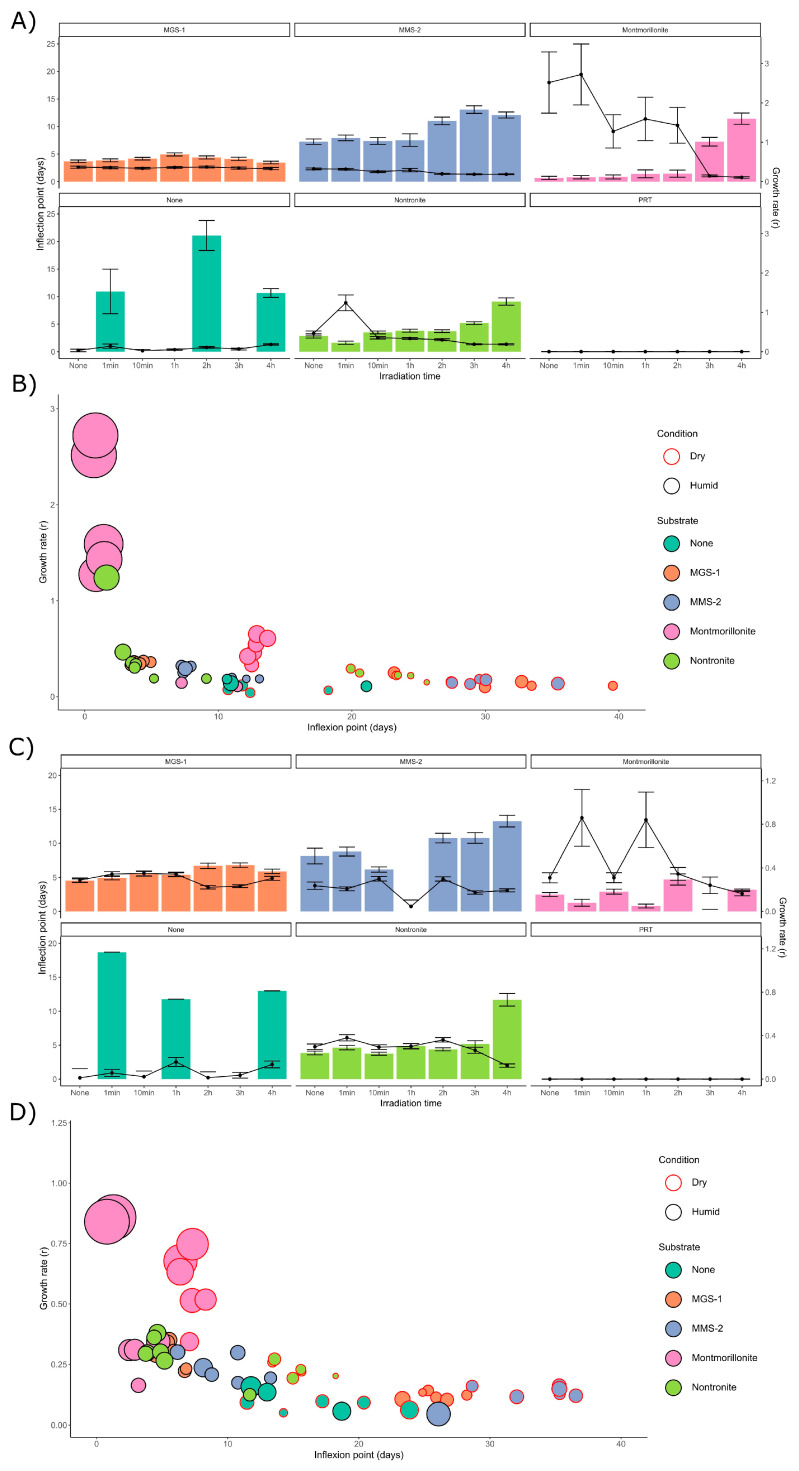
(**A**) Inflection point (in days) for the logistic regression obtained from modeling the growth of *D. muscorum* after being irradiated within a 1 mm thick layer of water in five mineral substrates. (**B**) Representation of the growth rate (r, points) and inflection day (t_mid, bars) of *D. muscorum*, irradiated either while dry or wet. Each point corresponds to the model derived from five replicates of each cyanobacterium, substrate and dose. As no significant differences took place in t_mid up to doses of 1320 kJ/m^2^, points are not differentiated based on irradiation time. The radius of each point is the standard error in growth rate (*y*-axis). (**C**,**D**) Equivalent data for *Desmonostoc* sp.

**Figure 8 microorganisms-13-01083-f008:**
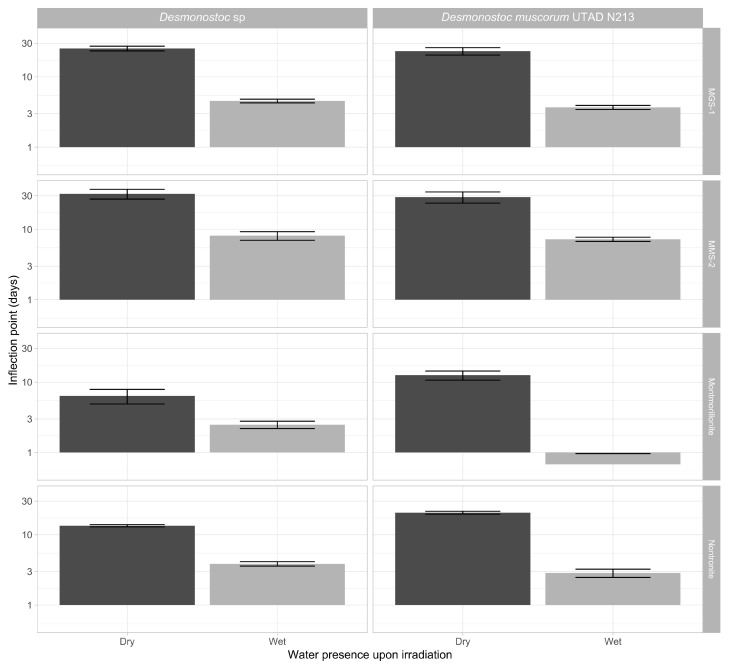
Variation in the inflection point (t_mid) of the two *Nostoc* species in the absence of UV-B radiation, in the presence of four mineral substrates, either desiccated or not. Each value corresponds to the t_mid derived from modeling five replicates of each cyanobacterium, substrate and dose. The y-axis depicts inflection point on a base 10 logarithmic scale.

**Table 1 microorganisms-13-01083-t001:** Assay of the oxidative pressure of each Martian regolith simulant through the consumption of DTT, a reducing agent. The absorbance was measured at 412 nm; SD stands for standard deviation; and the consumption was calculated in pmol DTT/g.

	Absorbance	SD	Consumption (pmol DTT/g)
Control+	0.906	0.091	0
MGS-1	0.412	0.043	545
MMS-2	0.284	0.032	686
PRT	0.232	0.034	744

## Data Availability

All datasets and R scripts used for statistical analysis and plotting of HPLC and Raman spectroscopic data in the study are available at Zenodo via 10.5281/zenodo.15342177 under a MIT license.

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
