# Peer review of "Survival of Filamentous Cyanobacteria Through Martian ISRU: Combined Effects of Desiccation and UV-B Radiation"

_microorganisms, 2025, doi:10.3390/microorganisms13051083_

Round 1

Reviewer 1 Report (Previous Reviewer 1)

Comments and Suggestions for Authors

Authors have improved this version of the document, in which the methods, tables and figures were highly enhanced to reflect the scientific signficance

Author Response

Line 123-134. Please revise this paragraph to contain the aims and objectives of the study rather than the results of the study. In this form, it is more suitable for the Conclusion than for the Introduction.

We have amended this section and removed all mentions to results.

Line 136. I think this section should be redesigned, as it contains not only methods but also results. It is logical to specify here that the determination was performed using molecular methods and morphology with the help of the indicated primers, microscopy and a guide (it should be noted that neither the type of microscope nor the guide is specified by the author). The result of the determination itself would be better placed in the Results section and provided with a table with the closest homologs and similarity percentages, and possibly a phylogenetic tree or microphotographs.

We have only left pure methodology here, and have moved the results of the 16S sequencing, along with new tables on the identification of the strains and a Maximum likelihood tree.

Line 139. Here and below indicate the strain number for Desmonostoc sp. too.

This strain in uncharacterized and not described and as such lacks a strain number.

Line 144. Please remove the brackets from the abbreviation of medium, as it causes misunderstanding.

Done.

Line 213. Here and below please add the word ‘medium’ to the abbreviation MBL, otherwise it sounds strange.

Done.

Figures 4-6. Please make all captions in black font, white and gray are hard to see.

Both figures have been amended.

Line 378, 442. Please correct the plural of 'media' to the singular 'medium'.

Done.

Line 571. Please explain more specifically what mechanisms we are talking about here. Have you considered scytonemin-poducing strains of cyanobacteria for participation in such experiments? Perhaps they will be more resistant to UV.

We have gone into more detail on this topic in the discussion.

Reviewer 2 Report (New Reviewer)

Comments and Suggestions for Authors

The authors have considered a very interesting topic on the potential use of cyanobacteria for the settlement of Mars. Experiments have shown the resistance of cyanobacteria to extreme conditions and the possibility of their growth on substrates similar to those on Mars. Small comments to improve the readers' understanding of the paper are given below.

Line 123-134. Please revise this paragraph to contain the aims and objectives of the study rather than the results of the study. In this form, it is more suitable for the Conclusion than for the Introduction.

Line 136. I think this section should be redesigned, as it contains not only methods but also results. It is logical to specify here that the determination was performed using molecular methods and morphology with the help of the indicated primers, microscopy and a guide (it should be noted that neither the type of microscope nor the guide is specified by the author). The result of the determination itself would be better placed in the Results section and provided with a table with the closest homologs and similarity percentages, and possibly a phylogenetic tree or microphotographs.

Line 139. Here and below indicate the strain number for Desmonostoc sp. too.

Line 144. Please remove the brackets from the abbreviation of medium, as it causes misunderstanding.

Line 213. Here and below please add the word ‘medium’ to the abbreviation MBL, otherwise it sounds strange.

Figures 4-6. Please make all captions in black font, white and gray are hard to see.

Line 378, 442. Please correct the plural of 'media' to the singular 'medium'.

Line 571. Please explain more specifically what mechanisms we are talking about here. Have you considered scytonemin-poducing strains of cyanobacteria for participation in such experiments? Perhaps they will be more resistant to UV.

Author Response

Line 123-134. Please revise this paragraph to contain the aims and objectives of the study rather than the results of the study. In this form, it is more suitable for the Conclusion than for the Introduction.

We have amended this section and removed all mentions to results.

Line 136. I think this section should be redesigned, as it contains not only methods but also results. It is logical to specify here that the determination was performed using molecular methods and morphology with the help of the indicated primers, microscopy and a guide (it should be noted that neither the type of microscope nor the guide is specified by the author). The result of the determination itself would be better placed in the Results section and provided with a table with the closest homologs and similarity percentages, and possibly a phylogenetic tree or microphotographs.

We have only left pure methodology here, and have moved the results of the 16S sequencing, along with new tables on the identification of the strains and a Maximum likelihood tree.

Line 139. Here and below indicate the strain number for Desmonostoc sp. too.

This strain in uncharacterized and not described and as such lacks a strain number.

Line 144. Please remove the brackets from the abbreviation of medium, as it causes misunderstanding.

Done.

Line 213. Here and below please add the word ‘medium’ to the abbreviation MBL, otherwise it sounds strange.

Done.

Figures 4-6. Please make all captions in black font, white and gray are hard to see.

Both figures have been amended.

Line 378, 442. Please correct the plural of 'media' to the singular 'medium'.

Done.

Line 571. Please explain more specifically what mechanisms we are talking about here. Have you considered scytonemin-poducing strains of cyanobacteria for participation in such experiments? Perhaps they will be more resistant to UV.

We have gone into more detail on this topic in the discussion.

This manuscript is a resubmission of an earlier submission. The following is a list of the peer review reports and author responses from that submission.

Round 1

Reviewer 1 Report

Comments and Suggestions for Authors

Abstract

please maintain the length at 200 words according to MDPI guidelines

INTRODUCTION

normally, introduction section has no subheadings. please try to unify the different sections into one single section

METHODS

please revise line 175: K2PO4, is it K2HPO4 or KH2PO4?

on line 242, is it UV-B or UV-C?

RESULTS

if possible try to enhance the quality of the figures

there are some minor issues with the units, some are wrongly written (without subindex)

after writing for the first time Nostoc muscorum, the second time and further on please change to  "N. muscorum" 

please describe how the data obtained can be compared with other cyanobacteria, specially with highly resistant strain such as thermophilic cyanobacteria

how do authors believe that environmental stress can damage or regulate the response of the strains, specifically on antioxidant production or DNA repair pathways?

The growth curves presented by the authors require a statistical analysis. since each curve fits a different model, the final growth values can be enhanced accordingly to their statistical significance

since most cyanobacteria grow well in a prior of 30-50 days. How long do the authors believe that those cyanobacteria can withstand those harsh conditions?

Reviewer 2 Report

Comments and Suggestions for Authors

Reviewers comments on article “Filamentous cyanobacteria survival using Martian ISRU: effects of desiccation and UV-B radiation”

General comments:

Lines 89-90 "On Earth, cyanobacterial oxygen production is substantial, with marine cyanobacteria of the genus Prochlorococcus alone accounting for about 5% of global photosynthesis" - this sentence needs reference. Also please mention other photosynthetic m.o. and their part in global photosynthesis.

Line 93 “thus making them more efficient” in what?

Line 98 please mention that Arthrospira platensis (Spirulina) is  accepted as food in EU along with C. vulgaris…

Please state wavelength ranges of UV-A and UV-B

Please provide latitude and longitude for Rio Tinto (Huelva) and Atacama desert…

Line 125 “shine as the prime candidates” please use more scientific expression

Line 134 Please state clearly here the novelty and impact of your paper

line 145 - “10 µmol/m2s” use 10 µmol/(m2s) or 10 µmolm-2s-1 (proper units)

Line 188 Something is wrong with the equation. Please correct this.

dH20 appears in many places in text please correct this

Line 205 - you mention that the growth was measured at OD 440 nm, but before you claimed to use OD 685 for growth (chlorophyll) and OD620 for blue pigment, please clarify your methods…

Line 225 – what is thrice-washed?

Line 235 the take – then taken?

Line 237 them taken - then taken?

Line 270 please state the unit of growth rate (r)

Table 1 looks very strange

Line 291 dH2O (represented by the red line with dots) and in the legend dH2O is represented by blue line + black dots. Please clarify

Figures 2,3,4,5 and 6 – can you please change the background from dark grey to white for better visibility?

Line 357 “The first ten or so days” so ten days or more or less? Please reformulate

Lines 360-361: This sentence belongs to the Discussion section

Figure 5. What happened with the “polvo del rio tinto - PRT” substrate? It doesn’t appear in the figures… Was it discarded?

Results section is too long and exhaustive, which can confuse readers. Please try to shorten it and make it more focused on the most important results. I have the same opinion for Discussion and Conclusions section, both should be modified to be more focused and informative with only relevant references, discussed and compared with your results.

Line 573 “Nontronite behaved worse than montmorillonite” please reformulate

Comments on the Quality of English Language

Language is sometimes confusing with wrong expressions like thrice-washed, the take and them taken instead of then taken etc. Also sometimes language should be more scientific and less beletristic...

I strongly recommend that the manuscript undergoes some English lecturing

Reviewer 3 Report

Comments and Suggestions for Authors

Based on the provided microscopy images, it is evident that these three strains do not belong to Nostoc or Anabaena cylindrica, indicating an incorrect identification. Please provide 16S sequence data or other relevant identification information to verify their classification. Due to this misidentification, further evaluation of the subsequent experiments is not possible at this stage.